# Assessing the Mechanism of Fluoxetine-Mediated CYP2D6 Inhibition

**DOI:** 10.3390/pharmaceutics13020148

**Published:** 2021-01-23

**Authors:** Malavika Deodhar, Sweilem B. Al Rihani, Lucy Darakjian, Jacques Turgeon, Veronique Michaud

**Affiliations:** 1Precision Pharmacotherapy Research and Development Institute, Tabula Rasa Health Care, Lake Nona, Orlando, FL 32827, USA; mdeodhar@trhc.com (M.D.); srihani@trhc.com (S.B.A.R.); ldarakjian@trhc.com (L.D.); jturgeon@trhc.com (J.T.); 2Faculty of pharmacy, Université de Montréal, Montréal, QC H3C 3J7, Canada

**Keywords:** fluoxetine, norfluoxetine, drug interactions, CYP2D6, metabolism, competitive inhibition, mechanism-based inhibition

## Abstract

Fluoxetine is still one of the most widely used antidepressants in the world. The drug is extensively metabolized by several cytochrome P450 (CYP450) enzymes and subjected to a myriad of CYP450-mediated drug interactions. In a multidrug regimen, preemptive mitigation of drug–drug interactions requires knowledge of fluoxetine actions on these CYP450 enzymes. The major metabolic pathway of fluoxetine leading to the formation of its active metabolite, norfluoxetine, is mediated by CYP2D6. Fluoxetine and norfluoxetine are strong affinity substrates of CYP2D6 and can inhibit, potentially through various mechanisms, the metabolism of other sensitive CYP2D6 substrates. Remarkably, fluoxetine-mediated CYP2D6 inhibition subsides long after fluoxetine first passes through the liver and even remains long after the discontinuation of the drug. Herein, we review pharmacokinetic and pharmacogenetic information to help us understand the mechanisms underlying the prolonged inhibition of CYP2D6 following fluoxetine administration. We propose that long-term inhibition of CYP2D6 is likely a result of competitive inhibition. This is due to strong affinity binding of fluoxetine and norfluoxetine to the enzyme and unbound fluoxetine and norfluoxetine levels circulating in the blood for a long period of time because of their long elimination half-life. Additionally, we describe that fluoxetine is a CYP2C9 substrate and a mechanism-based inhibitor of CYP2C19.

## 1. Introduction

Patients presenting with several chronic conditions such as psychiatric disorders, cardiovascular disorders, diabetes, and rheumatoid arthritis often require multiple therapeutic approaches, including the use of numerous drugs, to achieve clinical benefits and experience favorable outcomes [1,2,3,4]. The co-administration of multiple drugs required to treat complex conditions and associated comorbidities can lead to major pharmacokinetic and/or pharmacodynamic drug–drug interactions (DDIs) [5,6,7]. These circumstances also impact drug safety, adherence, and efficacy [8,9]. The cytochrome P450 (CYP450) monooxygenase superfamily of enzymes is commonly involved in the metabolic processes regulating drug disposition [10,11]. CYP450 enzymes are responsible for metabolizing about 75% of all prescribed medications and contribute greatly to DDIs in patients with polypharmacy [12].

In recent years, an increased incidence of depression with an associated increase in the use of antidepressants has been noted [13,14,15,16]. Fluoxetine is still one of the most widely used antidepressants worldwide, with 21,913,276 prescriptions filled in the United States in 2017 [17]. Fluoxetine is a selective serotonin reuptake inhibitor (SSRI) indicated for the treatment of psychiatric conditions such as major depressive disorder, obsessive compulsive disorder, bulimia nervosa, and panic disorder [18]. Eli Lilly received patent protection for fluoxetine in 1974, and the drug was approved in the USA for medical use in 1987 [19]. It is considered one of the most stimulating SSRIs; this property, along with its a rather favorable side-effect profile, offers clinical advantages [20,21]. It is well recognized that CYP2D6 is the major CYP450 enzyme involved in the metabolism of fluoxetine [22]. CYP2D6 is also known to be involved in the metabolism of opioid painkillers, such as tramadol and codeine, beta-blockers, class I antiarrhythmic drugs, first-generation H1-antagonists, tricyclic antidepressants, and other small amine-containing compounds [23,24,25]. Concomitant use of fluoxetine with any of these agents poses a risk for potential DDIs leading to therapeutic failure or adverse reactions. More importantly, inhibition of CYP2D6 by fluoxetine has been reported to subside long after fluoxetine first passes through the liver and even lingers long after the discontinuation of the drug [26,27]. In this review, we aim to shed light on the mechanisms of fluoxetine-mediated CYP2D6 inhibition and its interactions with other CYP450s (CYP2C9, CYP2C19 and CYP3A4). We discuss whether fluoxetine is either a perpetrator or victim drug and whether fluoxetine interactions with various CYP450s may lead to reversible or irreversible inhibition.

## 2. Mechanisms of CYP450 Inhibition

Drug interactions associated with CYP450 inhibition are classified as reversible (i.e., competitive or noncompetitive) or irreversible (i.e., mechanism-based inhibition) [28]. Firstly, competitive inhibition occurs when two substrates, present in the surrounding of the enzyme at the same time, compete for the same active site. Whether a substrate is a perpetrator or a victim drug is a function of its respective affinity for the binding site, its concentration in the proximity of the enzyme, and the basal level of the enzyme activity [28]. Under conditions of competitive inhibition, separating the time of administration has often been a successful interaction-mitigating strategy, since in this type of inhibition, the basal enzyme (protein) level is not impacted [28]. A noncompetitive inhibitor usually has no structural resemblance with the substrate of an enzyme; the noncompetitive inhibitor generally binds to an allosteric site leading to a conformational change, rendering the active site unavailable for substrate binding [29]. Lastly, mechanism-based inhibition is a subset of time-dependent inhibition, which can occur when a substrate forms a reactive metabolite or, in rare cases, a substrate binds so tightly to the enzyme acting as a suicide inhibitor, leading to an irreversible reduction of enzymatic activity [28,30,31]. In cases of noncompetitive inhibition and mechanism-based inhibition, separating administration time of the perpetrator and victim drugs cannot prevent the interaction [28].

## 3. Fluoxetine Metabolism 

Fluoxetine is extensively metabolized by various CYP450s such that only ~2.5% of the administered dose is excreted unchanged in urine [32]. The *N*-demethylation of fluoxetine produces norfluoxetine, an active metabolite with potent serotonin selective reuptake inhibition (SSRI) activity [21]. Other metabolic pathways include glucuronidation and *O*-dealkylation converting fluoxetine into *p*-trifluoromethylphenol which subsequently is transformed into hippuric acid, a glycine conjugate [33,34,35]. Fluoxetine metabolism to norfluoxetine is stereoselective and involves CYP2D6, CYP2C9, and possibly CYP2C19 (Figure 1). In vivo in humans, it is evident that CYP2D6 plays a much greater role than any other enzyme [22,36]. Since fluoxetine remains under various conditions a possible substrate of all these enzymes, the likelihood of DDIs when co-administered with other drugs is real. However, the extent or significance of these interactions depends on whether fluoxetine acts as a victim drug or perpetrator drug at each of these enzymes.

CYP2D6 is mainly responsible for metabolizing both S- and R-fluoxetine to norfluoxetine corresponding enantiomers (intrinsic clearance, CLint = 2.24 uL/min/pmol of CYP, and 2.91 uL/min/pmol of CYP, respectively) [37]. However, the preferential metabolism of S-fluoxetine to S-norfluoxetine has been demonstrated in pharmacogenetic studies [38].

## 4. Mechanism of CYP2D6 Inhibition by Fluoxetine

S-Fluoxetine and S-norfluoxetine are both substrates and exhibit a high binding affinity to CYP2D6 (Ki = 68 nM and 35 nM, respectively; Ki is reflective of the binding affinity of a potential inhibitor; the lower the Ki, the higher the binding affinity is and smaller the amount of the drug needed in order to inhibit the activity of that enzyme) [39]. Thus, it is clear that competitive inhibition with sensitive CYP2D6 substrates is expected, and that fluoxetine and norfluoxetine will likely act as perpetrator drugs for several other CYP2D6 substrates. Accordingly, fluoxetine administration for 8 days is associated with a significant decrease in dextromethorphan clearance (a sensitive CYP2D6 substrate) and metabolism to dextrorphan: a 9- to 18-fold increase in the dextromethorphan/dextrorphan ratio [26,40]. Remarkedly, however, as is discussed further below (see Section 5), it took two to three weeks after discontinuation of fluoxetine for CYP2D6 activity to return back to baseline [26,27]. Understanding the exact mechanism of this prolonged inhibition of CYP2D6 activity poses certain challenges.

The first possibility is that prolonged CYP2D6 inhibition by fluoxetine might be due to noncompetitive inhibition. As a CYP2D6 substrate, fluoxetine must bind to the active site to be metabolized into norfluoxetine. By definition, pure noncompetitive inhibitors are not metabolized by the enzyme they inhibit [28]. Additionally, the 3D-conformational structure of noncompetitive inhibitors generally does not resemble that of enzyme substrates. Since fluoxetine is a CYP2D6 substrate, it appears not to meet some of the criteria of a pure noncompetitive inhibitor.

Second, fluoxetine could be a mixed CYP2D6 inhibitor, i.e., exhibiting both competitive and noncompetitive inhibition. This mechanism of inhibition gives rise to very potent enzyme inhibitors, often acting as perpetrator drugs with other sensitive substrates, while their pharmacokinetics are not affected by other inhibitors of the same enzyme [41]. In vitro studies showed that quinidine (a potent CYP2D6 inhibitor) was capable of inhibiting the *N*-demethylation of fluoxetine with a magnitude similar to that observed with anti-CYP2D6 antibodies [37,42]. These data suggest that the probability of noncompetitive CYP2D6 inhibition by fluoxetine is rather unlikely but leave open the possibility for competitive inhibition.

Third, since fluoxetine is a CYP2D6 substrate, studies have been conducted to determine whether fluoxetine produces the time-dependent, mechanism-based inactivation of CYP2D6 [43]. Bertelsen et al. determined the time-dependent inhibitory potency of fluoxetine in human liver microsomes [43]. The IC_50_ for CYP2D6 inhibition by fluoxetine did not decrease with various times of pre-incubation periods to allow a potential reactive metabolite to be formed. These observations led to the conclusion that fluoxetine does not exhibit a time-dependent increase in its inhibitory potency and thus is not a mechanism-based inhibitor. Further evidence for the lack of noncompetitive and mechanism-based inhibition of CYP2D6 by fluoxetine is also obtained looking at results from pharmacogenetic studies. It is well known that poor metabolizers of CYP2D6 (carriers of two nonfunctional CYP2D6 alleles) exhibit decreased clearance characteristics of CYP2D6 probe substrates compared to normal CYP2D6 metabolizers (carriers of two functional CYP2D6 alleles) [44]. If noncompetitive or mechanism-based inhibition were to occur following administration of fluoxetine, the initial differences in fluoxetine pharmacokinetics observed between normal and poor metabolizers would disappear with time, as normal metabolizers would be phenoconverted to poor metabolizers due to the time-dependent decrease in CYP2D6 activity (phenoconversion being a phenomenon by which an genotype-predicted phenotype is transformed into another phenotype by factors such as drug interactions). In that sense, Fjordside et al. investigated the stereoselectivity of fluoxetine demethylation into norfluoxetine in patients on days 7, 14, and 23 following fluoxetine administration (20 mg) [38]. The plasma concentrations of fluoxetine and norfluoxetine enantiomers were measured in poor and normal metabolizers of CYP2D6. First, it was confirmed that CYP2D6-mediated demethylation of fluoxetine is stereoselective toward the S-enantiomer. Second, their results demonstrated a persistent difference between normal and poor metabolizers in the concentrations of S-fluoxetine and S-norfluoxetine even after 3 weeks of treatment [38]. Higher concentrations of S-fluoxetine and lower concentrations of S-norfluoxetine were observed in poor metabolizers compared to normal metabolizers. As indicated before, the persistent difference observed in fluoxetine pharmacokinetics between poor and normal metabolizers suggests a lack of CYP2D6 auto-inhibition via noncompetitive or mechanism-based inhibition. Scordo et al. also reported on the influence of CYP2D6 polymorphisms on fluoxetine steady-state pharmacokinetics: the median S-norfluoxetine/S-fluoxetine ratios were higher in homozygous compared to the heterozygous normal metabolizers for CYP2D6 [45]. Finally, a correlation between the fluoxetine/norfluoxetine ratio and CYP2D6 genotype (number of active *CYP2D6* genes) was also reported in other clinical studies [36,46,47,48]. Overall, these findings support that (1) fluoxetine is a substrate of CYP2D6, (2) fluoxetine is not a noncompetitive inhibitor of CYP2D6, (3) CYP2D6 regulates the concentration of the S-enantiomer and the formation of S-norfluoxetine even after multiple doses, and (4) CYP2D6 inhibition by fluoxetine is not time-dependent, and therefore incompatible with mechanism-based inhibition. Thus, by eliminating the alternatives, it appears that fluoxetine is a strong affinity substrate of CYP2D6 and that the reduced CYP2D6 activity is likely mediated through competitive inhibition at the active site. However, the prolonged time-course of CYP2D6 inhibition which subsides even after stopping fluoxetine administration requires further investigation, since it is not commonly encountered when dealing with competitive inhibitors.

## 5. Time Course of CYP2D6 Inhibition

Conceptually, separating the time of administration between two competing substrates, such that their maximum concentrations in the liver or plasma do not overlap, is a meaningful way to mitigate DDIs caused by competitive inhibition. For most drugs, CYP2D6 inhibition is observed following drug absorption during the hepatic first pass phase (CYP2D6 is not expressed in the intestinal wall) since drug concentrations in the portal vein (which irrigates perilobular hepatocytes) are much higher than concentrations observed after peak plasma concentrations when drugs return to the liver via the hepatic artery. Generally, the ratio of drug concentrations observed in the portal vein during the absorption phase to that measured in the hepatic artery after the peak plasma concentration (when the absorption phase is almost completed) is very large (likely in the range of 100 to 1000). This is why separating the time of administration of two substrates that compete for the same enzyme circumvents competitive inhibition between these two substrates. However, as mentioned previously, in the case of fluoxetine, a prolonged CYP2D6 inhibition over two to three weeks was observed when dextromethorphan was used as a probe substrate [26,27]. Other reports have suggested that these prolonged effects may advantage fluoxetine over other SSRIs in avoiding withdrawal syndromes [49,50]. Significant DDIs have also been reported between fluoxetine and tricyclic antidepressants, several of them also being CYP2D6 substrates [51]. Desipramine (50 mg) mean plasma concentrations were increased 4.4-fold when co-administered with 20 mg fluoxetine for 20 days [52]. Twenty-one days post-discontinuation of fluoxetine, mean plasma concentrations of desipramine still remained 2.4-fold higher than baseline, demonstrating the long-term impact of fluoxetine administration on other CYP2D6 substrates [52]. It is established that both fluoxetine and norfluoxetine have very long half-lives of 4–6 days and 4–16 days, respectively [18]. Additionally, it has been reported that during chronic dosing, the terminal half-life of fluoxetine and norfluoxetine might increase to become slightly longer (8 and 19.3 days, respectively) [53]. Some particular characteristics may contribute to the prolonged effects of fluoxetine administration on CYP2D6. Persistent high concentrations of unbound fluoxetine and norfluoxetine in plasma are one of these factors. Hence, the degree of in vivo interaction between an enzyme and a substrate can be estimated using the [I]u/Ki ratio, where [I]u refers to the concentration of the unbound drug and Ki is the in vitro inhibition constant for this enzyme [39]. Using Ki values of 68 nM and 35 nM for S-fluoxetine and S-norfluoxetine, Sager et al. reported a [I]u/Ki ratio of 5.8 for S-fluoxetine and of 4.5 for S-norfluoxetine (calculated using the unbound concentration in the liver), suggesting that the unbound concentration of these drugs at steady-state is much higher than the concentration required to block 50% of the CYP2D6 active site [39]. This concept can also be appreciated through the consideration of in vivo metabolic inhibition potential using pharmacokinetic parameters. Since the CYP2D6-mediated demethylation of fluoxetine to norfluoxetine is stereoselective, a more precise estimation is obtained by looking at S-enantiomer concentrations. Sager et al. studied the plasma levels of fluoxetine enantiomers following the administration of fluoxetine 60 mg for 12 days to healthy volunteers; the average plasma concentration of S-fluoxetine and S-norfluoxetine was 770 ± 270 nM and 320 ± 110 nM (mean ± SD), respectively [39]. The plasma fraction unbound (fu) was estimated at 0.14 and 0.13 for S-fluoxetine and S-norfluoxetine, respectively. Based on these parameters, the predicted unbound plasma concentrations of S-fluoxetine (108 nM) and S-norfluoxetine (42 nM) are within the range of CYP2D6 Ki values reported in different studies for S-fluoxetine (68–220 nM) and for S-norfluoxetine (35–310 nM). In these estimations, the average steady-state plasma concentrations instead of the maximal concentration were used to assess the potential metabolic inhibition of CYP2D6. The use of [I]u plasma/Ki can underestimate the prediction of in vivo drug interactions compared to the use of [I]in,u liver/Ki ([I]in,u being the maximal unbound hepatic inlet concentration) [54]. In brief, the long-lasting CYP2D6 competitive inhibition by S-fluoxetine and S-norfluoxetine could be explained by their unbound concentrations remaining much higher than their Ki value throughout a dosing interval and days thereafter.

## 6. CYP2C19 and Fluoxetine

Fluoxetine is a substrate (Km = 251 μM and 154 μM; Clint = 0.186 uL/min/pmol of CYP and 0.275 uL/min/pmol of CYP for the S- and R-enantiomers, respectively) and a mechanism-based inhibitor of CYP2C19, as CYP2C19 is marginally involved in metabolizing fluoxetine into norfluoxetine [34,48]. In addition to the *N*-demethylation pathway, fluoxetine and possibly norfluoxetine undergo CYP2C19-mediated *O*-dealkylation to form the *p*-trifluoromethylphenol metabolite [33,35]. A 30-min fluoxetine pre-incubation shows a time-dependent reduction in CYP2C19 activity in vitro [55]. Additionally, Stresser et al. suggested stereoselective inhibition demonstrating that S-fluoxetine is a more potent mechanism-based inhibitor of CYP2C19 than the R-enantiomer (the S-enantiomer exhibited a lower affinity for the inactivation for CYP2C19 but a faster rate of inactivation compared to the R-fluoxetine) [55]. In contrast, Sager et al. assessed the risk of the irreversible inhibition of CYP2C19, and based on their in vitro ratios, they found that R-fluoxetine and S-norfluoxetine were most likely contributing to the irreversible inhibition of CYP2C19 [39]. In agreement with this study, in vitro time-dependent prediction models suggested that in pooled human liver microsomes, the inhibition of CYP2C19 was similar with racemic–fluoxetine or racemic–norfluoxetine. When each enantiomer was considered independently, S-norfluoxetine was predicted to contribute the most to CYP2C19 inhibition [56]. Findings have been inconsistent; discrepancies between in vitro inhibition studies may be explained by enzyme sources, range of concentrations, and the in vitro models used. Overall, fluoxetine and norfluoxetine are likely to be perpetrator drugs for other CYP2C19 substrates. In support of this statement, the in vivo phase I clinical studies have shown reduced formation of the clopidogrel active metabolite (mediated largely by CYP2C19) and increased platelet aggregation when fluoxetine was co-administered with clopidogrel [57]. However, a study by Bykov et al. could not demonstrate a significant clinical impact on bleeding events [58]. 

## 7. CYP2C9 and Fluoxetine

In vitro studies suggest an important role for CYP2C9 in the metabolism of fluoxetine with stereoselectivity toward the R-enantiomer [37,46]. Pharmacogenetic studies evaluating fluoxetine transformation by microsomes from individuals with variant alleles coding for non- or decreased functional CYP2C9 activity suggested that fluoxetine is a substrate of CYP2C9 [36,59]. However, the affinity of fluoxetine toward CYP2C9 appears relatively low (Km value = 1660 μM and 922 μM for the S- and R-enantiomers, respectively), suggesting that it would likely be a victim drug if a stronger CYP2C9 substrates or inhibitors were co-administered [46]. In vivo, CYP2C9 seems to play a very minor role in the pharmacokinetics of fluoxetine [47]. These findings suggest that significant clinical drug interactions are unlikely through this pathway. There is no information available supporting the involvement of CYP2C9 in the sequential metabolism of norfluoxetine.

## 8. CYP3A4 and Fluoxetine

In vitro experiments have suggested a role for CYP3A4 in the metabolism of fluoxetine (Km value 33.5 μM; Clint = 0.316 uL/min/pmol of CYP) [37]. Furthermore, S-fluoxetine and R-norfluoxetine are reported to show some degree of mechanism-based inhibition towards CYP3A4 [56,60]. Through the use of pharmacokinetic-pharmacodynamic modeling, it has been predicted that a 60–62% reduction is possible in CYP3A4 activity when all enantiomers of parent drug and active metabolite are considered [60]. However, in vivo pharmacokinetic interaction studies with CYP3A4 sensitive substrates such as midazolam did not demonstrate the significant modification in midazolam pharmacokinetics as expected from in vitro studies [61]. In addition to a potential contribution to the *N*-demethylation pathway, in vitro studies suggested that CYP3A4 could contribute to the *O*-dealkylation of fluoxetine and perhaps norfluoxetine [33,35]. Based on in vitro studies, under conditions of nonfunctional CYP2C19 activity, it could be speculated that the impact of CYP3A4 on fluoxetine *O*-dealkylation can be revealed; however, the incongruity between in vivo and in vitro evidence will require further studies to provide any definitive conclusions.

## 9. Conclusions

Based on the information presented above, the strong CYP2D6 binding affinity of both S-fluoxetine and S-norfluoxetine, their unbound steady-state plasma concentrations above their respective Ki values, and their long elimination half-life all contribute to the prolonged inhibitory effects of fluoxetine and norfluoxetine on CYP2D6 activity. The mechanism of CYP2D6 inhibition by fluoxetine and norfluoxetine appears to be principally explained by competitive reversible inhibition. Clinically, separating the time of administration between fluoxetine and other CYP2D6 substrates will not be effective in mitigating DDIs, and the dose reduction of sensitive CYP2D6 substrates co-administered with fluoxetine is warranted to avoid adverse drug events.

## Figures and Tables

**Figure 1 pharmaceutics-13-00148-f001:**
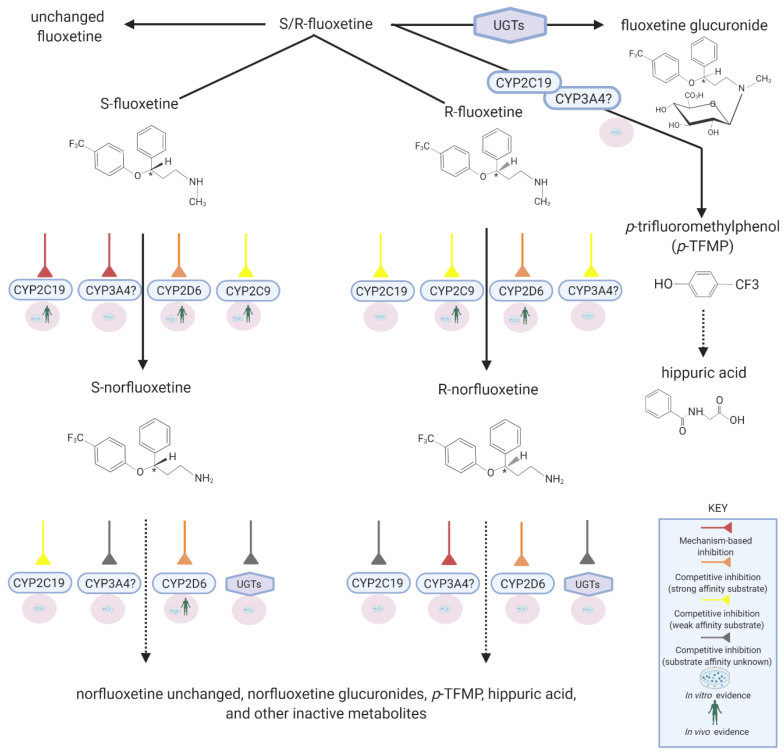
Disposition pathways of fluoxetine. Created with BioRender.com.

## Data Availability

Not applicable.

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
