# Peer review of "Assessing the Mechanism of Fluoxetine-Mediated CYP2D6 Inhibition"

_pharmaceutics, 2021, doi:10.3390/pharmaceutics13020148_

Round 1

Reviewer 1 Report

Line 59 - I think the verb should be - to "subside"

Figure 1 - I suggest that a question mark be place below the "CYP3A4" placements due to the incongruity between the in vitro and in vivo findings (Section 8).

Section 5 - lines 199 - 200 - Please indicate whether the variance for the two mean values is provided as SD or SEM.

Section 8.  - This is the weakest part of the whole manuscript.  The conclusion provided as speculation is not warranted at the present time.  I suggest that the conclusion should be that the potential role of CYP3A4 clinically in fluoxetine disposition remains to be demonstrated due to the incongruity between in vivo and in vitro metabolism data reported in the scientific literature.

References - For all references to the internet sources, the date of last successful acquisition should be provided.

Author Response

Reviewer 1 comments:

  1. Line 59 - I think the verb should be - to "subside":

The necessary change was made on line 59.

  1. Figure 1 - I suggest that a question mark be place below the "CYP3A4" placements due to the incongruity between the in vitro and in vivo findings (Section 8):

As suggested by the reviewer, the figure has been modified to place a ”?” where CYP3A4 mechanism-based inhibition is mentioned, i.e., for S-fluoxetine and R-norfluoxetine.

  1. Section 5 - lines 199 - 200 - Please indicate whether the variance for the two mean values is provided as SD or SEM:

These values are mean ± SD and this specification has been added to the text on line 200.

  1. Section 8. - This is the weakest part of the whole manuscript. The conclusion provided as speculation is not warranted at the present time. I suggest that the conclusion should be that the potential role of CYP3A4 clinically in fluoxetine disposition remains to be demonstrated due to the incongruity between in vivo and in vitro metabolism data reported in the scientific literature:

As suggested by the reviewer, the sentence was modified on lines 256-259 as follows:

“Based on in vitro studies, under conditions of non-functional CYP2C19 activity, it could be speculated that the impact of CYP3A4 on fluoxetine O-dealkylation can be revealed, however the incongruity between in vivo and in vitro evidence will require further studies to provide any definitive conclusions.”   

  1. References - For all references to the internet sources, the date of last successful acquisition should be provided.

References have been updated and dates are provided.

Reviewer 2 Report

Deodhar et al. prepared a manuscript “Assessing the Mechanism of Fluoxetine-Mediated CYP2D6 Inhibition”. This review is well thought out and present valuable source for understanding fluoxetine interaction with cytochrome P450 enzymes.

Remarks to consider:

  1. Isoform, isoenzyme although frequently found in literature are not recommended to be used with cythochromes P450. As these enzymes do not follow the concept of lock and key, it is advisable to refer to them as enzyme, as per well established nomenclature.
  2. Lines 75-77 “Lastly, mechanism-based inhibition could be time-dependent (when a substrate forms a reactive metabolite) or time-independent (when the substrate itself binds covalently to the enzyme).” This should be corrected: all mechanism based inhibitions are time dependent inhibitions, but all time dependent inhibitions are not mechanism or metabolism dependent. Although rare, time dependent inhibition can be just a result of slow binding to the enzyme (reference Drug Metab Dispos 42(9):1438-46. doi: 10.1124/dmd.114.059295).
  3. Figure 1. please add structural formulas of compounds on the scheme.
  4. Line 99. Km although often referenced as indicator of binding affinity is not per se a binding constant. Please reword this sentence: “S-Fluoxetine and S-norfluoxetine both exhibit high binding affinity to CYP2D6 (Km= 68 nM and 35 nM, respectively)”
  5. Line 190. Please verify values “Using Ki values of 68 nM and 35 nM for S-fluoxetine and S-norfluoxetine” as the seem to be same as Km values.
  6. Line 239. Same comment as for #4
  7. In whole text when referring to the name of reaction heteroatom should be written in italic e.g. N-demethylation, O-dealkylation

Author Response

Reviewer 2 comments:

  1. Isoform, isoenzyme although frequently found in literature are not recommended to be used with cytochromes P450. As these enzymes do not follow the concept of lock and key, it is advisable to refer to them as enzyme, as per well-established nomenclature:

As suggested by the reviewer, the term “isoform” has been changed to “enzyme” throughout the manuscript.

  1. Lines 75-77 “Lastly, mechanism-based inhibition could be time-dependent (when a substrate forms a reactive metabolite) or time-independent (when the substrate itself binds covalently to the enzyme).” This should be corrected: all mechanism-based inhibitions are time dependent inhibitions, but all time dependent inhibitions are not mechanism or metabolism dependent. Although rare, time dependent inhibition can be just a result of slow binding to the enzyme (reference Drug Metab Dispos 42(9):1438-46. doi: 10.1124/dmd.114.059295).:

We agree with the reviewer, mechanism-based inhibition is a subset of time-dependent inhibition which defines inactivation by a chemically reactive metabolite. We meant to refer to mechanism-based inhibition via suicide inhibition (direct action of substrate on enzyme), and alternate mechanism via reactive metabolite. To avoid confusion, the sentence has been modified to convey the proper message as follows:

“Lastly, mechanism-based inhibition is a subset of time-dependent inhibition—which can occur when a substrate forms a reactive metabolite or  in rare case, a substrate binds so tightly to the enzyme acting as a suicide inhibitor—leading to an irreversible reduction of enzymatic activity.”

  1. Figure 1. please add structural formulas of compounds on the scheme:

As suggested by the reviewer, structural formulas of compounds have been added on the Figure 1.

  1. Line 99. Km although often referenced as indicator of binding affinity is not per se a binding constant. Please reword this sentence: “S-Fluoxetine and S-norfluoxetine both exhibit high binding affinity to CYP2D6 (Km= 68 nM and 35 nM, respectively)”: 

We are using Ki as an indicator for binding affinity as a potential inhibitor towards CYP2D6 substrates and do not intend to mislead the reader to think it’s a binding constant. We have thus added an explanatory sentence on lines 103-104, as follows:

“S-Fluoxetine and S-norfluoxetine are both CYP2D6 substrate and exhibit high binding affinity to CYP2D6 (Ki= 68 nM and 35 nM, respectively; Ki is reflective of the binding affinity of a potential inhibitor, the lower the Ki, the higher is the binding affinity and smaller amount of the drug is needed in order to inhibit the activity of that enzyme).  

  1. Line 190. Please verify values “Using Ki values of 68 nM and 35 nM for S-fluoxetine and S-norfluoxetine” as the seem to be same as Km values.:

This sentence is appropriate, and it refer to the Ki values. The previous sentence (see comment #4 has been modified).

  1. Line 239. Same comment as for #4

This sentence is appropriate, and it refer to the Ki values. The previous sentence (see comment #4 has been modified).

  1. In whole text when referring to the name of reaction heteroatom should be written in italic e.g. N-demethylation, O-dealkylation:

These changes have been made throughout the manuscript.